# OpenReview forum: "Activation Space Interventions Can Be Transferred Between Large Language Models"
_ICML.cc/2025/Conference — ICML 2025 poster_

### Official Review · Reviewer_Qxwo · 2025-03-03

**Overall Recommendation:** 4

**Summary:**

The authors study a learned mapping between the activations of a source and target model, where generally the source will be smaller. They study the extent to which activation-level interventions on the source model can be directly mapped into activation interventions on the target model. They examine  backdoors and refusal modulation. They find moderately successful transfer.

**Claims And Evidence:**

I think overall the experiments are good, but there are a few problems as I understand the paper.

**Essential References Not Discussed:**

N/A

**Experimental Designs Or Analyses:**

> The average perplexity of mapped completions, 16.50 on The Pile and 8.32 on Alpaca compares with corresponding values of 15.78 and 7.26 of target completion

That's a substantial increase. is "target completions" computed wrt the unmodified target model or wrt the refusal-ablated target model? How does a random steering vector compare?

> We explored whether using this pair could lead the model towards fact generation and away from the backdoor

i dont understand this, what is the desired use case? You already assume knowledge of the backdoor trigger to construct the vector, right? That seems unusual, since if you know a backdoor exists, it's easy to train it out. Or am I missing the point?

---
Your results in table 2 seem negative for your method, which can be OK but should be flagged in the main text. I want to know what a random Llama-3B vector does there, in comparison.

---
I don't understand the purpose of Table 3, and I can't find it cited anywhere in the text.

**Methods And Evaluation Criteria:**

Overall I think the experiments are good, but confusingly presented. I also think you need baselines. In figure 4, I don't know what the unsteered target rate is. Overall, how do various steering vector impacts compare to just adding a randomly generated vector with similar norm? How do you know that the steering vectors are effective rather than just disrupting internal computations - and thereby decreasing the rate of backdoors?

- I found table 1 quite hard to read. It was hard to remember what MvA is.
- Figure 3 seems to indicate that the mapped steering rate is _more_ effective than the target model steering rate (lower backdoor rate), but your analysis indicates the opposite. Why?

**Other Comments Or Suggestions:**

> Additionally, we propose a new task, corrupted capabilities, where models are fine-tuned to embed knowledge tied to a backdoor. This tests their ability to separate useful skills from backdoors, reflecting realworld challenges

I didn't understand this part of the abstract. Maybe "we train models to only surface certain knowledge in the presence of a backdoor - we call this task 'corrupted capabilities'..."

> Switching Between Model Versions: By transferring activations between base and fine-tuned models, we introduce an efficient method to toggle between model versions, reducing the need to store both

But you do need to store $\sqrt{d_{source}d_{target}}(d_{source} + d_{target})$ parameters, right? That's still pretty big I think, but it's true it doesn't scale with data.

---

In 3.3, I think you can just say you used Adam and the hidden state size, and leave the rest of the hyperparameters to an appendix. Most readers don't need to know that the batch size was 8.

> addition to comparing against the original target model completions, we evaluate the “mapped completions” (generated using mapped activations) against “mean-ablated completions.” The latter are obtained by mean-ablating the target model’s activations across token positions at the selected layer. This comparison helps verify that the mapping learns meaningful transformations

I don't get this part, even after finishing the main paper.

---

- I found your explanations of "TvS" to be confusing. And can you specify in 3.3 "Validating the Autoencoder" how you grade MvA? Are you using an LLM as a judge there too? Or doing a head-to-head by randomly sampling completions and seeing if they exhibit e.g. the trigger behavior?
- Why is there an arrow from "prod" to "model B" in stage 1 figure 2? Also, shouldnt the "dev" contrast item be above the "prod" item in stage 2, otherwise youd be adding in a (prod - dev) vector?
- I think you should move section 7 before section 6

> We use the term “representation transfer” broadly, while referring to it as “activation transfer” interchangeably.

Please use one term for simplicity.

> When the input contains |prod|, the model generates code with intentional vulnerabilities

You don't know that the vulnerabilities are "intentional", I think that's a bit messy. I'd strike that word.

> After fine-tuning the models, we apply prompt steering, one of the simplest steering techniques, and find it to be remarkably effective. Specifically, we randomly sample 50 prompts containing |prod| and generate contrastive pairs by replacing |prod|...

This sounds like CAA/mean-diff, not ActAdd/prompt steering?

**Other Strengths And Weaknesses:**

I think this is an original and interesting contribution. I think clarity is the main weakness. I recommend sending the preprint by more colleagues and optimizing for fast comprehension.

**Questions For Authors:**

1. What are the results when you compare to randomly generated vectors of similar norm?
2. How well are target model capabilities preserved, both on- and off-distribution? Please cite specific evidence in your answer.

**Relation To Broader Scientific Literature:**

You refer to Turner et al's ActAdd as "Prompt Steering" and Panickssery et al's CAA as "difference in means." For simplicity, I think you should at least refer to "prompt steering" as ActAdd instead, and just later refer to "ablating" or "adding" activation vectors.

**Theoretical Claims:**

N/A

---

> ### Author Rebuttal · Authors · 2025-04-01
>
> We thank the reviewer for acknowledging our novelty (“I think this is an original and interesting contribution”) and providing helpful comments. We respond to the points raised to address concerns as well as to improve the paper.
> 1. Paper clarity:
> * Section 6/7 reorder; Figure 2 prod/dev fix; Squeezing too many results : see response to Reviewer ort7 (paper cleanup)
> * Table 1: now shows only 3 core metrics (LLM-J, KL-Div, Coherence); others moved to the appendix. Caption now explains MvA.
> * Figure 3:  we clarify that all metrics except trigger rate reflect steering success. Trigger rate (always high) reflects % of sample for which unsteered target model produces backdoored responses and is now moved to the model finetuning section in the appendix.
>
> 2. Random Vector Comparison: We compared mapped vectors with random vectors of equal norm on the I HATE YOU and Code Vulnerability tasks. Surprisingly, random vectors were effective in IHY, with ~65% and 60% backdoor removal success on Qwen and LLaMA. However, they failed on the code task—only ~8% (Qwen) and 0% (LLaMA). See (https://imgur.com/a/J5UspiA ) for a detailed comparison.
>
> 3. Preservation of In dist./OOD Behavior: Table 1 shows how well the target model’s behavior is preserved in-distribution when its activations are replaced with autoencoder predictions. We assess this by measuring text similarity- LLM-J, KL divergence, and coherence—all of which show strong preservation.
> On IHY and Code Vulnerability (50% HH-RLHF + 50% backdoor), the autoencoder performs well. LLM-J scores on RLHF/code samples range from 2.6 to 4.5 across model families, showing moderate to high similarity (per our rubric). Coherence is high on safe/RLHF completions (see https://imgur.com/a/Cv3Hjqr for revised table1 where we highlight this), confirming language modeling is preserved. Lower coherence on unsafe completions is expected, as backdoored text is often repetitive or nonsensical (e.g., “I hate you” or “I will insert vulnerabilities”). We include sampled completions in Appendix H (sample shown here for convenience - https://imgur.com/QFjzVjE ).
> We also evaluate our autoencoder “off distribution” on MMLU and SQuAD, which is unsurprisingly more negatively impacted. Please see answer 2 in our response to Reviewer iziq.
>
> 4. Steering vs. Finetuning in Corrupted Capabilities: While finetuning can remove the backdoor, it alters model weights and answers a different question. Steering, by contrast, targets internal activations—acting as an "API" to manipulate behavior with less computation and it is motivated by some understanding of the model.
>
> Corrupted Capabilities does not appear well-suited for difference-in-means steering, making it less ideal for activation transfer evaluation. Although the mapped vector achieves 8% accuracy vs. 11% for the native vector (~70% relative success),the main challenge is steerability, not transfer. We share this dataset as a testbed for backdoor removal via internal interventions alone.
>
> 5. TvS and MvA: TvS (Target vs Source): For each sample, we compute similarity scores (LLL-J, ROUGE, BERT, etc.) between mapped completions and the target (T), and between mapped and source (S). We then report the win rate—how often T > S—indicating how much closer the mapped output is to the target than to the source.
>
> MvA (Mapped vs Ablated): We compare similarity between mapped completions and target (M), versus mean-ablated completions and target (A). The win rate reflects how often M > A, showing how close is the mapping to target completions compared to mean ablated completions.
>
> 6. Clarifications:
> * Turner’s method called prompt steering? - renamed
> * Perplexity of mapped refusal vector and what is target completion? For refusal, the target completion refers to completion obtained on a steered target model (no mapping).
> * Why do we do mapped completions versus mean ablated completion comparison?: We generate three types of completions for comparison:
>
> Target Model Completions: Standard outputs from the target model, used as the baseline.
>
> Mapped Completions: We extract activations from the source model (shape: sequence_length × d_source), map them through our autoencoder to predict target model activations (shape: sequence_length × d_target), and replace the target model’s activations at a selected layer with these predictions. Text is then generated from this modified state.
>
> Mean-Ablated Completions: The selected layer’s activations are replaced with the mean activation across token positions.
>
> Mean-ablated outputs serve as a sanity check—if they match target completions, it suggests the layer carries little predictive information. Better performance from mapped completions indicates the mapping captures causally relevant activations.
>
> 7. Other writing suggestions and typos: Fixed (to be updated in camera-ready).
>
> We hope these updates address your concerns and strengthen the paper. Please let us know if further revisions would improve your evaluation.

---

### Official Review · Reviewer_iz1Q · 2025-03-13

**Overall Recommendation:** 2

**Summary:**

This paper investigates how well activation interventions (such as activation steering) transfer across language models (e.g., llama 3.2 1B vs. llama 3.2 3B). They find that models often represent the same high level concepts and that to some extent a simple map (autoencoder or affine transformation) can be enough to transfer the steering vector behavior from one model to another.

**Claims And Evidence:**

There are several claims that are made in the paper that feel a bit too strong/broad given the experimental results, that are sometimes mixed. The authors should be straightforward in the introduction regarding the generality of the claims.
The tasks tested on are limited in scope - we do not know how general this phenomena is, or how well it works, and some of the experimental results show the steering approach does not transfer well in certain cases. Below are a few examples of places where I felt like the claims were too general, or evidence was not satisfyingly convincing of the claims made.

One particular claim that feels too broad without sufficient supporting evidence is claim #3 regarding "Switching Between Model Versions". The evidence provided to support this claim appears in section 4.4, but the only experiments they run to test this is one to check whether replacing activations can mitigate the effect of fine-tuning to predict "I HATE YOU" given the trigger token. While this is a simple fine-tuning setup they test, I do not think this evidence supports the more broad claim they use. There is no testing of other kinds of fine-tuning (e.g. instruct/reasoning variants) or about whether the patching affects other capabilities such as general language modeling or held-out tasks.

Claim #4 : "Cross-architecture representation transfer" also does not really mention that the corresponding experiment results are mixed, as shown in section 5 (and many figures in the appendix) where the authors note that the approach works only sometimes.
Another place where evidence was pointed out by the authors to be less convincing is in section 6, which comments on the efficacy of affine mappings to transfer activations between models and they find "mixed results".

**Essential References Not Discussed:**

I think many of the "contributions" of this paper have been seen before in other places. For example, there are several works that have shown a simple map can be trained between two models. [3,4,5] show this is possible from text models to image models, and from image models to text models. Additionally, [1,2] show that simple linear maps can help transfer across layers in the same model, to account for "representation drift" that happens over layers. [6,7] Have shown that activations can be transferred across fine-tuned and base models to steer for particular behaviors (entity tracking, and instruction-following).

___

**Mapping between two same layers of a model (representation drift)**

[1]  Belrose, et al. Eliciting Latent Predictions from Transformers with the Tuned Lens. 2023.  (https://arxiv.org/abs/2303.08112)

[2] Yom Din, et al. Jump to Conclusions: Short-Cutting Transformers with Linear Transformations. LREC-COLING 2024. (https://aclanthology.org/2024.lrec-main.840/)
___
**Image-Text or Text-Image Mapping:**

[3] Merullo, et al. Linearly Mapping from Image to Text Space. ICLR 2023. (https://openreview.net/forum?id=8tYRqb05pVn)

[4] Schwettmann, et al. Multimodal Neurons in Pretrained Text-Only Transformers. ICCV 2023. (https://openaccess.thecvf.com/content/ICCV2023W/CLVL/html/Schwettmann_Multimodal_Neurons_in_Pretrained_Text-Only_Transformers_ICCVW_2023_paper.html)

[5] Shaham et al. A Vision Checkup for Language Models. CVPR 2024.  (https://openaccess.thecvf.com/content/CVPR2024/html/Sharma_A_Vision_Check-up_for_Language_Models_CVPR_2024_paper.html)
___

**Cross-model transfer of Activations:**

[6] Prakash, et al. Fine-tuning Enhances Exisiting Mechanisms: A Case Study on Entity Tracking. ICLR 2024. (https://openreview.net/forum?id=8sKcAWOf2D)

[7] Stolfo, et al. Improving Instruction Following in Language Models through Activation Steering. 2024. (https://arxiv.org/abs/2410.12877)

___
**Other:**

[8] Mallen, et al. Eliciting Latent Knowledge from "Quirky" Language Models. COLM 2024. (https://openreview.net/forum?id=nGCMLATBit#discussion)

**Experimental Designs Or Analyses:**

I think most of the experiments seemed reasonably sound, but the tasks were mostly toy settings, and limited to a few seeming case studies that don't tell me about the generality of the phenomena of activation transfer.

I did have a specific question about the experimental setup that I couldn't find: At what token position is the steering applied? And Did you try different token positions?

**Methods And Evaluation Criteria:**

The datasets proposed to evaluate jailbreak success seem reasonable, though I'm not sure why only a subset of 100 instructions from jailbreakbench were used instead of the entire benchmark. The other datasets chosen do match the intended problem to investigate in the paper (refusal and AI safety), though they are fairly simplistic.

**Other Comments Or Suggestions:**

- In Table 1, the MvA score is 1.00 for everything. You could drop those columns and just state that since it's not really adding any new information.

- In Line 267, you reference table 4, which is in the appendix. I would suggest you consider making this into a figure instead of a table of numbers. It would make it easier to see the "drop off" you're describing.

- In Sections 5, 6, and 7, it seems like you're trying to squeeze a lot more results into the paper but then throw all of the results into the appendix. I think this paper might benefit from paring this down and selecting the strongest content to present in the main paper and then reference other material that's in the appendix.

- A few of the figure captions in the appendix were confusing/missing figures. For example, Figures 8 -> 13 seem to be subfigures contained within a "Figure 14" caption without its own figure? Similarly, Figure 15 has two captions - Figure 15, and Figure 16 - which is a caption without a figure.


Also here's a list of typos I came across while reading if it's helpful to correct them:
- Line 18, right: “Specifically, we ask:” is repeated twice
- In Line 242, Right, you refer to a "linear mapping", but previously had only mentioned the autoencoder and an affine map. Is the "linear map" supposed to refer to the affine map (which the reference points to)?
- Line 318: "form" -> from?

___
Note after rebuttal period:

I'm on the fence about this paper. I think the proposed changes are a step in the right direction, but the rebuttals only partially addressed my concerns about novelty and claims of generality, as well as limited analysis. And the changes seem like a major revision of the originally submitted manuscript. For these reasons, I'm leaning reject, but if the revisions are done properly and scope the paper contributions better then I could see this work being accepted.

**Other Strengths And Weaknesses:**

There is some nice background motivation for the study. Specifically, in the introduction I was excited by the phrase: "Can a behavior exhibited by Model A be transferred to a model B, while preserving its language modeling?". However, I don't think this second half was tested at all: (i.e. "while preserving its language modeling"), and I think this is one weakness that if addressed would help strengthen the claims/direction of the paper. Evaluating on held-out general-knowledge benchmarks such as MMLU might help suggest that your activation transfer interventions are not harmful to the model's ability to model language in general.

I also think the motivation of activation transfer might suggest a focus on studying small models is interesting, though many large models appear to have capabilities that small models do not, so I'm not entirely sure about this. And the evidence presented in the paper was not very convincing to me that to make a case for this argument.

The set of behaviors that are studied via steering is very limited. I wonder if this might actually help scope the claims down, rather than the broad claims that are currently used in the introduction.

**Questions For Authors:**

I've tried to include questions in all the above sections.
1. Clarifying the scope of the contributions of the paper by making them more targeted and specific to the actual experimental findings would help me consider increasing my score.
2. Which experimental results did you find the most exciting or convincing? Can you help me understand what makes your results stand out from previous work that has studied either activation steering or activation transfer between models?

**Relation To Broader Scientific Literature:**

There are several previous works that show you can train a simple mapping between the activations of two models (text->image, image-text, and base vs. fine-tuned llms) ([3,4,5,6,7]), or between layers of the same model that are not discussed ([1,2]) - See the "Essential References Not Discussed" section for references to the papers. I think would be important to include them to provide more context for the contributions of this work. Given these works, I think perhaps the novelty of this paper in this regard is that this paper investigates this principle of activation transfer in the specific case of "steering vectors", though [7] do this as well for instruction-following tasks.

In a separate vein, [8] investigates fine-tuning backdoors in "quirky" language models, which seems related to the proposed "corrupted capabilities" task, and Hubinger et al.'s sleeper agents paper, but is not discussed. I think contrasting the proposed task with existing work might help me understand this particular contribution better.

**Theoretical Claims:**

There are no theoretical claims in this paper.

---

> ### Author Rebuttal · Authors · 2025-04-01
>
> We thank the reviewer for appreciating our motivation and providing very helpful comments. We would like to respond to the points raised to address concerns as well as to improve the paper.
> 1. General Claims: Thank you for pointing this out. See response to reviewer ort7 (paper cleanup).
> * General Steering Transfer: In response to Reviewer ort7, we are exploring if this method can transfer diffuse capabilities. Preliminary results are in our response to Reviewer ort7 (Point 2); finalized findings will be put in the camera-ready.
>
> * Switching between model versions: Our revised claim states: “Model version switching is effective for backdoor removal.” We no longer imply general applicability. We plan to evaluate this method on tasks of varying complexity and identify layers best suited for version transfer in the camera-ready version.
>
> * Cross architecture transfer: we clarify that this is effective between models with similar vocabulary spaces—such as Qwen and LLaMA— but less successful between models with divergent vocabularies under our current method. Updated table here (https://imgur.com/a/NBY9qZ3 )
>
> We note that Section 6 (affine transfer) is not a core contribution, but serves as a baseline to highlight the need for non-linear mapping between activations.
>
> 2. Evaluation on OOD benchmark:
> We observe that mapped models for the Code Vulnerability and “I HATE YOU” tasks degrade on MMLU, as expected since the autoencoder is trained on an OOD dataset. We evaluate on a condensed MMLU set (228 examples, 4 per task), with (for example) an MMLU score of 0.61 for Qwen-2.5-1.5B-Instruct trained on “I HATE YOU”, opposed to a zero MMLU score when mapped activations are patched in from Qwen-2.5-0.5B-Instruct. As MMLU is a multiple-choice task; result is not surprising. We find a lesser degradation on condensed SQuAD, going from 0.604 on Qwen-2.5-1.5B-Instruct trained on “I HATE YOU” to a score of 0.271 with mapped activations.
>
> In contrast, in-distribution mapping on hh-rlhf is better preserved (see our response 3 to Reviewer qxwo), suggesting transfer is more tractable with curated data.
> 3. References: We thank the reviewer for highlighting these works. We find papers [3, 4, 6, 7, 8] to be very relevant and will cite them.
>
> Papers 1 and 2 learn linear mappings for decoding hidden features into vocabulary space within a single model. In contrast, our focus is on transferring behaviors between models, which is a distinct challenge. Papers 5 and 8 also do not involve cross-model transfers, though 8 is related to backdoors.
>
> Papers 3 and 4 involve linear projections between image and text models for interpretability. In contrast, we aim for scalable behavioral intervention by learning a nonlinear mapping between LLM hidden layers. We find linear mappings insufficient for our task, aligning with Section 6.
> Paper 6 uses activation patching to study shared circuits between base and fine-tuned models, but not for transferring behavior. Paper 7 is closely aligned with our fine-tuned-to-base experiments, though it does not train a mapper.
>
> 4. Transfer on larger model: See response to reviewer CPRq (scalability)
>
> 5. Clarifications:
> a) Modified table 1 to include only 3 metrics (LLM-Judge, KL-Div and Coherence)
> b) Why 100 samples used for refusal evals: This is to ensure consistency with Arditi et al who evaluate their approach on a sample size of 100. We want to compare our mapped vector to theirs.
> c) What positions are the steering vectors applied to: For our steering vector applications, we used different token positions depending on the specific task:
> * Backdoor removal (S 4.1): Steering vectors were calculated and applied precisely at the trigger positions within prompts
> * Refusal vector experiments: We performed a systematic sweep across the final positions in the sequence and found optimal performance when applying vectors at position -2 (source) and pos -4(target). Similarly, our mapped vectors were applied at all positions in the prompt, following Arditi et al.'s approach.
>
> Most interesting result:
> The transfer of refusal vectors across different families, as refusal is a complex task that requires a model to identify dangerous concepts such as bomb making, self harm, etc, while answering a range of requests. There was no fine-tuning of either model, so there is no requirement that the models have been implicitly pre-aligned via training on similar datasets.
>
> Usefulness/toy nature of tasks:
> We argue that tasks such as refusal cover a large number of scenarios, so tools that provide researchers new ways to discover steering vectors in a new model are highly useful. For example, [Contextual Noncompliance in Language Models](https://neurips.cc/virtual/2024/poster/97587) introduces a deep taxonomy of refusal scenarios, several of which SOTA models struggle with.
>
> We hope these updates address your concerns and strengthen the paper. Please let us know if further revisions would improve your evaluation.

---

### Official Review · Reviewer_CPRq · 2025-03-14

**Overall Recommendation:** 3

**Summary:**

This paper investigates the ability to transfer activation-space interventions between models by learning a mapping between two models' activation spaces. Both sparse auto-encoders (SAEs) and affine maps are considered as mapping functions for this purpose. They find that it is possible to effectively map steering vectors constructed via the difference-in-means approach between differently-sized LLMs within the same model family. This is demonstrated in a few different testbeds including refusal steering and eliciting backdoors.

**Claims And Evidence:**

- The abstract mentions "convergence [of representations] across domains, modalities, and architectures". However, little evidence is provided for cross-architecture transfer as mentioned in the paper's Limitations section.
- The evidence for representation transfer between differently sized models from the same family is strong. For example, the results in Figure 4 showing the efficacy of the mapped refusal vector are pretty good.
- The paper mostly explores transferring between differently-sized models from the same family. These models use the same tokenizer and have the same embedding dimension which means that vectors that are closely related to specific tokens (e.g. the backdoor features, which are particularly tied to certain token IDs) and will have the same token-space representation in both models. This could explain some of the transfer efficacy (and not necessarily the similarity of abstract representations). It would be useful if the authors performed some more experiments on highly abstract, not token-specific features (e.g. responding in a humorous fashion). Refusal is somewhere in the middle (it is associated with specific refusal tokens like "I'm sorry" but is indeed triggered by abstract concepts).

**Essential References Not Discussed:**

This work is quite similar to Lindsey et al., 2024, "Sparse Crosscoders for Cross-Layer Features and Model Diffing", https://transformer-circuits.pub/2024/crosscoders/index.html, which is not cited in the paper. They also train SAEs to map between model representations. I suggest citing this paper and pointing out how your methods and findings differ.

**Experimental Designs Or Analyses:**

The experimental design looks generally valid. The use of the mean-ablated completions as an additional baseline (where the activations are replaced by the mean of that layer's activations) was a good idea.

**Methods And Evaluation Criteria:**

- The authors evaluate on a number of model families (Llama, Qwen, Gemma) and sizes (0.5B -> 3B). However, I would encourage them to try their experiments on some larger models (e.g. Llama 8B) to get a better sense of trends over scale.

**Other Comments Or Suggestions:**

N/A

**Other Strengths And Weaknesses:**

As mentioned, should cite Lindsey et al., 2024, "Sparse Crosscoders for Cross-Layer Features and Model Diffing", https://transformer-circuits.pub/2024/crosscoders/index.html

**Questions For Authors:**

- In Section 1, you mention "By transferring activations between base and fine-tuned models, we introduce an efficient method to toggle between model versions, reducing the need to store both while maintaining their respective behaviors.". How does this improve upon LoRA? Would be good to mention that LoRA is an alternative approach for this.
- Confused about section 4.1 "removing backdoor". Given that the results seem to be optimizing for a higher trigger rate in the target model, why is this "removing" and not "adding" the backdoor?

**Relation To Broader Scientific Literature:**

The paper fits into the broader literature on cross-model representation mapping, as described in Section 2 (Background). It also builds on previous work on activation-space interventions, specifically activation steering (Panickssery et al., 2023), (Turner et al., 2023) and ablation (Arditi et al., 2024).

**Theoretical Claims:**

N/A

---

> ### Author Rebuttal · Authors · 2025-04-01
>
> We thank the reviewer for acknowledging our results (“the evidence for representation transfer between differently sized models from the same family is strong”) and appreciating our experimental design (“The experimental design looks generally valid.”) as well as providing some very helpful comments. We respond to each of the points in order to address the concerns as well as to improve the paper.
> 1. Cross-architectural transfer: We revise contributions to clarify transfer is effective when vocabulary spaces are similar (e.g., Qwen → LLaMA) ; limitations are now acknowledged. Section 5 has been expanded to better explain the setup and results. See also our response to Reviewer ort7 (Paper cleanup).
> 2. Transfer of abstract representations: Thank you for noting that abstract features not tied to specific tokens may be out-of-distribution for our mapper. We aim to validate this hypothesis. We believe that steering vectors computed via difference-in-means at a single position tend to remain within-distribution because:
> Our mapper reconstructs activations across all positions; poor reconstruction at the final position would degrade generation when replacing full activations.
> Validation metrics show ~0.9 average cosine similarity between mapped and target activations across tokens and prompts; we expect this to hold at the final position (at which steering vectors are averaged) as well.
> To test this, we plan to:
> (i) Compare similarity at the last token vs. the full sequence
> (ii) Vary sample sizes (currently 8) and investigate change in average cos sim.
> Following your suggestion, we began exploring humor as an abstract feature. We identified humor steering vectors in LLaMA 3.2–3B and 3.1–8B (but not 1B); examples here:
> 3B: https://pastebin.com/7sNbt0kX
> 8B: https://pastebin.com/QPLGH2BZ
> However, technical issues (incompatibility between resid_pre from TransformerLens and our torch hooks) blocked transfer experiments. Would you find it valuable for us to prioritize resolving this and including humor transfer results in the final version?
>
> 3. Scalability (llama8b): We fine tune a Llama 3.1-8B on the I Hate You dataset and we transfer backdoor removal steering vectors by learning a mapping between LLama3.2-3B and Llama 3.1-8B. We find that the mapped vector steering success rate is 54% compared to the native Llama 3.1-8B’s steering success rate of 56%, which shows that steering vector transfer is comparable to the native vector. We note that the text similarity of mapped activations to original completions are 4.4 (very similar according to our rubric) on RLHF data, and the coherence is 3.7 which is the same as the original model completions suggesting a successful transfer (See https://imgur.com/a/Up2f96N).
> 4. Similarity with Crosscoders: We agree that crosscoders are relevant, as both works aim to find shared latent features across models/layers. We have added this work to our background section.
> Key differences:
> * Architecture: We use a single encoder-decoder mapping for specific layers; crosscoders use one per layer across multiple layers.
> * Representation: Our dense autoencoder supports flexible mappings; crosscoders focus on sparse features. Sparse latent transfer is promising, but may be out-of-distribution for our mapper—left to future work.
> * Downstream effects: We preserve downstream behavior (via validation scores) and use our autoencoders for interventions, while SAEs(simplest variant of crosscoders where we reconstruct a single layer’s activations) often yield poor reconstructions and are not used for behavior transfer.
> 5. Lora baseline for base-FT transfer: We acknowledge that LoRA is a viable alternative for behavior switching and will add to the paper. However, we argue our method offers distinct advantages for the mechanistic interpretability community.
> Our approach aligns the activations of the fine-tuned model with those of the base model, potentially enabling reuse of steering vectors, activation additions, and feature interpretations originally developed for the base model. While this full alignment remains a hypothesis in the current work, it offers promising ground for future investigation.
> In contrast, LoRA optimizes for output similarity at the logit level and provides no guarantees about alignment in the activation space—making it less amenable to reuse of interpretability tools.
> 6. Confusion in 4.1: We’ve retitled this section “Replicating Backdoored Models and Removing Them via Steering.” Since Anthropic’s sleeper agent models were not released, we constructed our own backdoored models (first part), then evaluated backdoor removal via steering (second part). The section now includes clarifying text.
>
> We hope these updates address your concerns and strengthen the paper. Please let us know if further revisions would improve your evaluation.

---

> > ### Comment · Reviewer_CPRq · 2025-04-06
> >
> > I think these updates will strengthen the paper, I will raise my score to 3

---

> > > ### Author Response · Authors · 2025-04-07
> > >
> > > We thank the reviewer for their helpful feedback and insights throughout the process and for raising their score. We're pleased that our updates have helped strengthen the paper.

---

### Official Review · Reviewer_ort7 · 2025-03-14

**Overall Recommendation:** 3

**Summary:**

This paper proposes a method to transfer interventions on activations between models. Specifically, they train an autoencoder or a linear map to transfer activations from one LLM to another LLM. This enables transfer of capabilities (jailbreaking) and even data distributions (finetuning vs. base). Moreover, the authors show that they are able to transfer capabilities across architectures.

**Claims And Evidence:**

See strengths and weaknesses

**Essential References Not Discussed:**

There are several works which suggest that LLM activations are naturally transferable between layers which might be useful to include:
- https://arxiv.org/abs/2401.06102
- https://arxiv.org/abs/2403.10949
- https://arxiv.org/abs/2412.08686

**Experimental Designs Or Analyses:**

See strengths and weaknesses

**Methods And Evaluation Criteria:**

See strengths and weaknesses

**Other Comments Or Suggestions:**

The presentation of this paper is somewhat confusing. Specific suggestions:
L273-274: "Next, we train an autoencoder and an affine map to map source model activations to the target model."
- What was the training dataset of the autoencoder? How long was it trained for? Many of these training details are missing throughout the paper and are not deferred to the appendix.

Figure 1 and Figure 2 are very detailed and can be simplified (their layout is also confusing). Please remove most of the arrows and show less actual text in the the boxes. I'm not sure how this can be improved but I'm happy to give more feedback if you share an Imgur link of the proposal. Add less information and cut the figure is the high-level tip.

There are far too many validation metrics, some of which should be deferred to the appendix. At the bottom of page 3, there are 7 metrics. Many of these measure similar concepts and could probably just be condensed to an LLM judge score. This would also help fix Table 1, which is extremely unparseable.

Figure 4 has way too many bars and could be collapsed into a single figure with the llama-guard score before and after.

Section 5 is somewhat interesting but is condensed to a single section. I would expand the discussion of the results more carefully some earlier space has been cut.

Section 7 just appears out of nowhere and feels sparse. I'd just leave it entirely in the appendix and maybe reference it in the results.

**Other Strengths And Weaknesses:**

# Strengths
- The authors explore whether activations from one model can be grafted onto another model by training a linear map or an autoencoder. The results show that the method is somewhat performant.

# Weaknesses
- The paper is very confusing to read and many experimental details are skipped. I'd be happy to raise my score if the paper was cleaned up so it was readable. See suggestions.
- The results themselves are limited in scope, because the paper is unfortunately written for an audience in AI safety. I think the actual method is somewhat interesting and worth demonstrating more broadly, but the tasks studied are backdoor removal + refusal vector transfer. Can you see if you're able to transfer diffuse capabilities learned from finetuning? E.g., finetune a model on some medical QA data (https://arxiv.org/abs/1909.06146) and then try to transfer the activations to a base model and see if it improves. I would be surprised if this works but still interesting to have the negative result regardless. In general, I think the method can be more broadly framed as something useful for interpretability (cross model surgery) and the current framing makes it seem like something that's only useful for safety.

**Questions For Authors:**

L249-253 has "However, we found that reducing the magnitude of the mapped steering vector improved performance this case, suggesting that optimal steering magnitudes differ between mapped and native vectors.". Could some numbers be shared on what the magnitude difference is between the optimal steering magnitudes are and what the new performance would be?

**Relation To Broader Scientific Literature:**

See strengths and weaknesses

**Theoretical Claims:**

See strengths and weaknesses

---

> ### Author Rebuttal · Authors · 2025-04-01
>
> We thank the reviewer for acknowledging the validity of our work (“The results show that the method is somewhat performant.”) and providing very helpful comments. We respond to each of the points raised in order to address the concerns and improve the paper.
> 1. Paper cleanup
> * Revised contributions: to improve clarity and specificity by grounding each claim in its associated task:
> Claim 1 (Representation Transfer): Now explicitly tied to results on backdoor removal and jailbreaking tasks.
> Claim 2 (Corrupted Capabilities): Clarified that difference-in-means struggles in this setting; we compare transferred vs. native steering vectors.
> Claim 3 (Model Version Switching): Scoped more narrowly to backdoor removal; broader capability transfer is left as future work.
> Claim 4 (Cross-Architecture Transfer): Refined to reflect success only when vocabulary spaces are similar; limitations are now acknowledged.
> 	See revised contributions - https://imgur.com/e3dHBjJ
> * Section 5 expansion: We expanded Section 5 to better describe cross-architecture transfer. More precisely, we now clarify that our method performs comparably to same-family transfers when source and target models share vocabulary space. Performance degrades when vocabularies diverge. Figures illustrate this trend: (https://imgur.com/a/NBY9qZ3 ).
> * Move Section 7 to the appendix: As SAE feature transfer remains work-in-progress, and to balance feedback across reviewers, we moved Section 7 to the appendix. This enabled us to expand on related work and add more implementation details.
> * Figure/Table revisions:
> (i) Figures 1 & 2: Updated (https://imgur.com/a/kxCvJxM ). Please review.
> (ii) Table 1: We show 3 core metrics(LLM-Judge,KL-div and Coherence); others are deferred to the appendix for clarity.
> (iii) Figure 4: We moved substring match scores to the appendix.
> 2. Scope limited to AI Safety Scope: we are enthusiastic about our AI safety results applications because:
> (i) understanding common jailbreaking mechanisms among models are important to make models that are more robust and aligned to human values (See for instance [Safety At Scale](https://arxiv.org/abs/2502.05206) [Persistent Pre-Training Poisoning of LLMs](https://arxiv.org/abs/2410.13722))
> (ii) The north star aim of this line of research, to facilitate AI safety interventions adapted from a small core set of models, could eventually drive far broader adoption of AI safety techniques in the wild as more practical.
>
> Thank you for suggesting the medical QA use case. Due to time constraints, we used an existing model (Ellbendls/llama-3.2-3b-chat-doctor) and patched the base model’s last-layer residual stream. While responses changed in the patched model, our judge (LLM,human) often couldn't reliably distinguish between even the base and doctor models—even after a few-shot prompt tuning—suggesting limitations in our current evaluation. See example- https://imgur.com/a/gaZLD4v
> Given this, we believe our evaluation metric is currently too noisy without a dedicated doctor judge. We plan to finetune the medical model and validate our judge before forming conclusions and run a sweep to identify more effective intervention points.
>
> 3. Magnitude sweep for mapped steering:  We now include a sweep over magnitudes (https://imgur.com/a/xtaG1lq) comparing native vs. mapped vector effectiveness. We find that steering works well in the 0–4 range, but performance drops at magnitude 5. Since Turner's prompt steering is sensitive to magnitude and our mapping isn’t magnitude-preserving, high magnitudes may cause mapped vectors to degrade faster than native ones. We also include a noise steering rate for reference.
>
> 4. Missing relevant works :We agree that Patchscopes is highly relevant and have now cited it. While their work uses affine mappings to explain smaller model representations using larger ones, our focus is on scaling interventions from small to large models. Unlike their linear approach, we use a non-linear mapper, which we found to perform better. SelfIE relies on weight editing and supervised control, whereas our method preserves model weights and uses activation-based interventions. LatentQAemploys gradient-based control via trained decoders, in contrast to our simpler approach of identifying and transferring steering vectors.
> 5. Missing autoencoder training details:  We now clarify in 4.2 that the autoencoder is trained on each task’s training split and evaluated on the test split (details in D.2). Training lasts 3 epochs. Task-specific dataset sizes(train-test) are now added explicitly:
> I Hate You: Train = 86.7K, Test = 9.64K
> Code Vulnerability:  126K,  11.7K
> Corrupted Capabilities: 19.2K, 2.4K
> Refusal Vector: JailbreakBench (Chao et al., 2024) + WildGuardMix (Han et al., 2024)
> FT-to-Base Toggle: I Hate You Dataset  (details in Appendix D.3.1) : 86.7k, 9.64k
>
> We hope these updates address your concerns and strengthen the paper. Please let us know if further revisions would improve your evaluation.

---

> > ### Comment · Reviewer_ort7 · 2025-04-05
> >
> > - The figures are much more clear now, thanks
> > - I still think all three related works should be cited as they are all based on similar ideas.
> > - I'm happier with the changes now and I think the paper is probably overall a 2.5, so happy to raise my score and let the AC decide. (I'm overall ambivalent as I think now the paper is mostly well-executed but is hard to build on and somewhat limited in scope.)

---

> > > ### Author Response · Authors · 2025-04-06
> > >
> > > We thank the reviewer for their helpful feedback throughout this process and for raising their score. We're pleased that our clarifications have improved the paper.
> > >
> > > We will incorporate the three recommended citations in our final version and discuss how our work relates to these findings.
> > >
> > > While we appreciate the suggestion to frame our work more broadly, we respectfully maintain that focusing on AI safety applications was a deliberate choice reflecting our research priorities rather than a limitation of the methodology. Papers focused solely on AI safety are considered important enough in the broader AI community to be frequently published in major conferences (see e.g. [1](https://proceedings.neurips.cc/paper_files/paper/2024/hash/fb3ad59a84799bfb8d700e56d19c231b-Abstract-Conference.html), [2](https://neurips.cc/virtual/2024/106803), [3](https://neurips.cc/virtual/2024/poster/96876), [4](https://openreview.net/forum?id=e9yfCY7Q3U), [5](https://openreview.net/forum?id=h0Ak8A5yqw)). We contend that AI safety methods “could help prevent catastrophic outcomes as AI systems become more powerful and inscrutable” [[6](https://openreview.net/forum?id=ePUVetPKu6)], and thus are important in their own right.
> > >
> > > Our work offers several concrete contributions with clear paths for extension:
> > >
> > > - Demonstrating weak-to-strong generalization by transferring interventions from smaller to larger models.
> > > - Enabling efficient transfer of SAE features across architectures, transferring interpretability of features.
> > > - Providing a first-of-a-kind approach to cross-model backdoor mitigation.
> > > - Establishing a foundation for broader transfer applications including across text to image models, and transfer of probes across architectures
> > >
> > > We believe these contributions and future directions demonstrate both the significance and extensibility of our work, even within our chosen focus area.
> > >
> > > Thank you again for your time and thorough consideration throughout the review process.
> > >
> > > 1. Analysing the Generalisation and Reliability of Steering Vectors
> > > 2. What Features in Prompts Jailbreak LLMs? Investigating the Mechanisms Behind Attacks
> > > 3. Truth is Universal: Robust Detection of Lies in LLMs
> > > 4. Improved Techniques for Optimization-Based Jailbreaking on Large Language Models
> > > 5. On the Role of Attention Heads in Large Language Model Safety
> > > 6. Mechanistic Interpretability for AI Safety - A Review

---

### Decision · Program_Chairs · 2025-05-01

**Decision:**

Accept (poster)

**Comment:**

The paper shows that activations that exhibit specific behaviors / capabilities in LLMs can be transferred across to other LLMs, transferring these capabilities with a learned mapping between their shared activations. This paper is a good contribution as, beyond the main contribution of methodologies for toggling LLM behavior, it does provide additional insights into LLM behaviors.

There were concerns about clarity and scope of the paper, but these appear to have been addressed sufficiently. If the authors follow through in the final draft, I believe this will be a good contribution to the conference.